# The Portfolio Diet and HbA1c in Adults Living with Type 2 Diabetes Mellitus: A Patient-Level Pooled Analysis of Two Randomized Dietary Trials

**DOI:** 10.3390/nu16172817

**Published:** 2024-08-23

**Authors:** Meaghan E. Kavanagh, Songhee Back, Victoria Chen, Andrea J. Glenn, Gabrielle Viscardi, Zeinab Houshialsadat, John L. Sievenpiper, Cyril W. C. Kendall, David J. A. Jenkins, Laura Chiavaroli

**Affiliations:** 1Department of Nutritional Sciences, Temerty Faculty of Medicine, University of Toronto, Toronto, ON M5S 1A8, Canada; meaghan.kavanagh@utoronto.ca (M.E.K.); songhee.back@mail.utoronto.ca (S.B.); tori.chen@mail.utoronto.ca (V.C.); andrea.glenn@utoronto.ca (A.J.G.); gabrielle.viscardi@mail.utoronto.ca (G.V.); zeinab.houshialsadat@mail.utoronto.ca (Z.H.); john.sievenpiper@utoronto.ca (J.L.S.); cyril.kendall@utoronto.ca (C.W.C.K.); david.jenkins@utoronto.ca (D.J.A.J.); 2Toronto 3D Knowledge Synthesis and Clinical Trials Unit, Clinical Nutrition and Risk Factor Modification Center, St. Michael’s Hospital, Unity Health Toronto, Toronto, ON M5B 1W8, Canada; 3Department of Nutrition, Harvard TH Chan School of Public Health, Boston, MA 02115, USA; 4Li Ka Shing Knowledge Institute, St. Michael’s Hospital, Unity Health Toronto, Toronto, ON M5B 1W8, Canada; 5Division of Endocrinology and Metabolism, St. Michael’s Hospital, Unity Health Toronto, Toronto, ON M5B 1W8, Canada; 6Department of Medicine, Temerty Faculty of Medicine, University of Toronto, Toronto, ON M5S 1A8, Canada; 7College of Pharmacy and Nutrition, University of Saskatchewan, Saskatoon, SK S7N 5E5, Canada

**Keywords:** Portfolio Diet, dietary portfolio, diabetes, glycemic index, glycaemic index, glucose control, glycemia

## Abstract

**(1) Background:** The Portfolio Diet, a dietary pattern of cholesterol-lowering foods, is also rich in low glycemic index (GI) foods. While strong evidence supports clinically meaningful reductions in cholesterol, evidence on the relationship between the Portfolio Diet and diabetes management is lacking. **(2) Objective:** To evaluate the relationship between the Portfolio Diet and glycated hemoglobin (HbA1c) as a determinant of glycemic control among adults living with type 2 diabetes mellitus (T2DM). **(3) Methods:** Patient-level data was pooled from two randomized dietary trials of low glycemic index interventions compared to high cereal fibre control diets in adults living with T2DM where HbA1c was collected (clinicaltrials.gov identifiers: NCT00438698, NCT00438698). Dietary exposure was assessed using weighed 7-day diet records. Adherence to the Portfolio Diet and its pillars (nuts and seeds, plant protein, viscous fibre, plant sterols, monounsaturated fatty acid [MUFA] oils) was determined using the validated clinical Portfolio Diet Score (c-PDS). Multiple linear regression was used to assess the association between change in the c-PDS and change in HbA1c over 6-months with covariate adjustments. **(4) Results:** A total of 267 participants, predominantly White (67%) and male (63%), were included, with a mean ± standard error age of 62 ± 0.5 years, baseline BMI of 30.2 ± 0.3 kg/m^2^, HbA1c of 7.08 ± 0.03%, and a c-PDS of 4.1 ± 0.3 points out of 25. Change in the c-PDS was significantly associated with a change in HbA1c (β: −0.04% per point, 95% CI: −0.07, −0.02, *p* = 0.001). A 7.5-point (30%) increase in the c-PDS was associated with a 0.3% reduction in HbA1c. Of the individual pillars, a 1-point change in nut and seeds intake (β: −0.07%, 95% CI: −0.12, −0.02, *p* = 0.009) or in plant protein intake (β: −0.11%, 95% CI: −0.18, −0.03, *p* = 0.009) was associated with a change in HbA1c. Further analysis of plant protein intake revealed that an increase in dietary pulse intake, a particularly low-GI food, was significantly associated with a reduction in HbA1c (β: −0.24% per 1-cup points cooked pulses (226 g) or 2 c-PDS points, 95% CI: −0.45, −0.03, *p* = 0.028). **(5) Conclusions:** Among adults living with T2DM, the Portfolio Diet was associated with lower HbA1c over a 6-month period, predominantly driven by two pillars: nuts and seeds and plant protein, particularly dietary pulses. These data have implications for including the Portfolio Diet in dietary recommendations for glycemic control in T2DM. A trial demonstrating the direct causal effect of the Portfolio Diet in a diverse group is warranted.

## 1. Introduction

Globally, the substantial increase in the prevalence of type 2 diabetes mellitus (T2DM) incidence [1,2] has led to a significant burden on healthcare systems [3]. In Canada, the direct healthcare cost of diabetes is estimated to reach $39 billion by 2028 [4]. To address the need for management strategies, clinical practice guidelines recommend nutrition therapies as the cornerstone of care given their ability to reduce excess weight and improve glycemic control [5,6]. The Portfolio Diet, a therapeutic cholesterol-lowering dietary pattern, is recommended by diabetes, cardiovascular and obesity clinical practice guidelines internationally [6,7,8,9,10,11,12,13,14]. The Portfolio Diet includes five pillars: nuts and seeds, plant protein, viscous fibre, plant sterols, and monounsaturated fatty acid (MUFA) oils. Randomized controlled trials have shown the Portfolio Diet results in clinically meaningful reductions in low-density lipoprotein-cholesterol (LDL-C), equivalent to statin medication [15], as well as other established cardiovascular risk factors [16]. Prospective cohort studies have shown that the Portfolio Diet is associated with improvements in glycemic control [17] and incidence of T2DM [18] and cardiovascular disease (CVD) [19].

The Portfolio Diet is rich in low glycemic index (GI) foods. Of its five pillars, the plant protein pillar encourages consumption of dietary pulses and soy foods, and the viscous fibre pillar encourages oats, barley, viscous fibre-rich breads, and temperate climate fruit [20]. The GI is a marker of carbohydrate quality that provides a physiological basis for ranking carbohydrate foods based on their ability to raise postprandial blood glucose levels [6]. Low-GI dietary patterns are recognized in diabetes and cardiovascular clinical practice guidelines internationally [6,8,21,22,23,24], and have been shown to reduce the incidence of T2DM and coronary heart disease in observational studies [25,26,27,28] and to improve both hemoglobin A1c (HbA1c) and CVD risk factors in randomized controlled trials [29].

Despite the Portfolio Diet’s demonstrated efficacy in lowering LDL-C and cardiovascular risk factors, benefits on glycemic control have only been examined in observational cohort studies thus far. We therefore investigated the effect of the Portfolio Diet in T2DM by analysing patient-level data from two completed randomized controlled trials comparing low GI diets to high cereal fibre diets in adults living with T2DM, Diabetes Management with a low Glycemic Index diet (DMGI), and DM-Magnet Residence Imagining (DM-MRI). As the Portfolio Diet is comprised of many low-GI foods, it was hypothesized that the Portfolio Diet would be inversely associated with HbA1c in adult participants living with T2DM. The objective of the present study was to assess the association between adherence to the Portfolio Diet and change in HbA1c in an exploratory analysis of two completed trials on low-GI diets in T2DM.

## 2. Materials and Methods

### 2.1. Study Design 

We pooled individual patient-level data from two randomized dietary trials in adults living with T2DM where change in HbA1c was assessed (clinicaltrials.gov identifiers: NCT00438698, NCT00438698). In these trials, adults were randomized to receive either a low-GI or a high-cereal fibre diet for either 6 months or 3 years (DMGI and DMGI-MRI, respectively). All participants were provided advice to follow the National Cholesterol Education Program Adult Treatment Panel III diet, with <7% saturated fat and <200 mg of dietary cholesterol a day. The designs and findings of each study have been published [30,31,32]. 

The low-GI diets included the dietary advice to consume low-GI foods, including oats, barley, oat bran and psyllium breads, peas, beans, chickpeas and lentils, pasta, and vegetables and fruit, particularly eggplant and okra and temperate climate fruit. The trials were approved by the research ethics boards of St Michael’s Hospital and the University of Toronto, Toronto, ON, Canada, and written consent was obtained from all participants. 

### 2.2. Participants 

Both trials included men or postmenopausal women living with T2DM (HbA1c ≥ 6.5 to ≤8.0%) taking oral antihyperglycemic agents at a stable dose for ≥2 (DM-MRI) or ≥3 months (DMGI). Figure 1 presents the participant flow from both trials. Dietary and HbA1c data were available at baseline and 6 months in a total of 267 participants. 

### 2.3. The Portfolio Diet 

The Portfolio Diet is a “portfolio” of key foods with established cholesterol-lowering properties [15]. The diet includes: 45 g of nuts and seeds (almonds, walnuts, pumpkin seeds, peanut butter, etc.), 20 g of viscous fibre (from psyllium, oats, oatbran, barley, eggplant, okra, apples, oranges, etc.), 50 g of plant protein (from tofu, soymilk, plant-based meat alternatives, beans, chickpeas, lentils, etc.), 2 g of plant sterols (from fortified foods and supplements), and 45 g of plant-derived MUFA (from extra virgin olive oil, cold pressed canola and soybean oil, avocado fruit, etc.) [20]. These gram targets are for a 2000 kcal diet. In addition to these 5 categories, the Portfolio Diet also has a background diet of <7% of total energy from saturated fat and <200 mg/day of cholesterol, in alignment with the National Cholesterol Education Program (NCEP) II diet [33]. 

### 2.4. Assessment of the Portfolio Diet

Dietary intake was collected using weighed 7-day diet records (7DDR) at baseline and 6 months. 7DDRs were entered into an ESHA Food Processor SQL (version 10.1.1; ESHA, Salem, OR, USA). Adherence to the Portfolio Diet and its pillars was determined using the validated clinical Portfolio Diet Score (c-PDS), ranging from 0 to 25 points [34]. The c-PDS is a pre-defined scoring method developed for use in clinical practice and trial settings as part of a digital translational tool (PortfolioDiet.app) of the Canadian Cardiovascular Society guidelines for cardiovascular risk reduction [7,35]. As this analysis included participants within a dietary trial setting using 7DDR, and the results are meant to directly inform the PortfolioDiet.app, the c-PDS was used. 

Appendix A presents the c-PDS pillar targets for a 2000-kcal diet. The c-PDS codes food items from the 7DDR and then classifies them into the five distinct pillars: nuts and seeds, plant protein, viscous fibre, plant sterols, and high MUFA oils. For each pillar of the c-PDS, points range from a minimum of 0 points to a maximum of 5 points (total score ranged from 0 to 25 points). For example, a ½ cup of chickpeas = 1 point of plant protein for a 2000-kcal diet. 

### 2.5. Outcome 

The primary outcome of this study was a change in HbA1c from baseline to 6 months. Fasting blood samples were analyzed by St. Michael’s Hospital’s routine analytical laboratory [30,31,32]. 

Exploratory outcomes included the associated change for other cardiovascular risk factors including blood lipids (LDL-C, non-high density lipoprotein cholesterol (non-HDL-C), total cholesterol, triglycerides, HDL-C, and blood pressure (systolic blood pressure [SBP] and diastolic blood pressure [DBP]). LDL-C was only calculated in persons who had triglycerides below 800 mg/dL in accordance with the NIH equation [36]. Non-HDL-C was calculated by subtracting HDL-C from total cholesterol.

### 2.6. Statistical Analyses 

The control and treatment arms of both trials were pooled together. The participants’ baseline characteristics are presented as means and standard errors for continuous variables and as the number and proportion for categorical variables. Baseline characteristics between treatment groups were assessed via two-sample t-tests for continuous variables and chi-square tests for categorical variables. Multiple linear regression was used to assess the association between change in the c-PDS, its pillars, and dietary pulses, and change in HbA1c over 6 months. Appendix A presents the conceptual model for the Portfolio Diet and HbA1c in adults living with T2DM. Adjustments included age (continuous), sex (female, male), smoking (yes, no), body mass index (BMI) (continuous), prior CVD event (yes, no), antihyperglycemic medications (yes, no), blood pressure medications (yes, no), lipid-lowering medications (yes, no), and trial (DMGI, DM-MRI) as covariates. Given that the c-PDS is already adjusted for energy, a separate adjustment for energy was not performed, as this would have been an overadjustment. 

Dietary GI was predicted to be a mediator for the Portfolio Diet and HbA1c. A mediation analysis for dietary GI was performed where change in dietary GI was included as a covariate. To evaluate the linear correlation between change in the c-PDS and change in GI, Pearson correlation coefficients were used. Also, to explore the role of body weight, a mediation analysis for body weight was also performed, similarly to dietary GI. Subgroup analyses for sex (female, male) and ethnicity (White, South Asian, East Asian, and other) were also investigated. All analyses were conducted using STATA version 17.0 (StatCorp, College Station, TX, USA). Statistical significance was set at *p* < 0.05.

## 3. Results

### 3.1. Participant Characteristics 

Table 1 presents the baseline characteristics of included participants by trial. A total of 267 participants were included in the analysis from the DMGI (*n* = 124) and DM-MRI (*n* = 143) trials. All participants had been diagnosed with T2DM and had an HbA1c between 6.5 and 8.0% and 6.5 and 8.5% at randomization in the DMGI and DM-MRI trials, respectively. Participants were 62 ± 0.5 years of age (mean ± standard error), predominantly male (63%) and White (67%) or South Asian (16%), with a baseline BMI of 30.2 ± 0.3 kg/m^2^, HbA1c of 7.08 ± 0.03%, and a c-PDS of 4.1 ± 0.3 points (out of 25 points). Based on BMI classifications for obesity (≥30 kg/m^2^), about half of the participants (48%) were considered obese. Appendix A presents pooled raw values at the trial’s baseline and end (6 months) for cardiometabolic risk factors at baseline and end (6 months), for control (high fibre dietary advice), and test (low glycemic index dietary advice) groups.

### 3.2. Portfolio Diet Adherence 

Table 2 presents the intakes of the Portfolio Diet pillars at baseline and 6 months, and change over the 6-month period in the pooled cohort of all participants from intervention and control groups in both trials. Pooled adherence to the Portfolio Diet in all participants increased from (mean ± standard error) 4.10 ± 0.17 points at baseline to 4.44 ± 0.18 at 6 months, corresponding to a 16% and 18% adherence. Of the five pillars, nuts and seeds (−0.27 ± 0.09) and plant protein (0.25 ± 0.06) had the largest changes over the 6 months, whereas viscous fibre had a small increase (0.15 ± 0.08) and plant sterols (0.00 ± 0.02) and MUFA oils (−0.01 ± 0.09) had no or minimal change. Change in total dietary pulses was also assessed in g/d, increasing by 18.1 ± 5.9 g at 6 months. 

Appendix A presents the change in the c-PDS from baseline to 6 months in the control (high fibre dietary advice) and test (low GI dietary advice) groups. The mean change for those in the control groups was −1.5 points, ranging from −7.8 to 7.35 points. The mean change in the c-PDS for those in the test groups was 1.7, ranging from −6.3 to 9.4. Appendix A presents the change in terms of individual c-PDS pillars and dietary pulses. 

### 3.3. Portfolio Diet and HbA1c

Table 3 and Figure 2 present the multiple linear regression analysis of the association between change in the c-PDS and change in HbA1c. An increase in the c-PDS was associated with a reduction in HbA1c (β: −0.04% per c-PDS point, 95% CI: −0.07, −0.02, *p* = 0.001) after adjustments for covariates, which corresponds to a 0.3% reduction in HbA1c per 7.5-point (30%) increase in the c-PDS. An increase in intake of nuts and seeds and plant protein was associated with a reduction in HbA1c by 0.07% (−0.12, −0.02), *p* = 0.009, and 0.11% (−0.18, −0.03), *p* = 0.009, per c-PDS point, respectively. There was no significant association for the other three Portfolio Diet pillars (*p* > 0.05). When looking at a key component of plant protein—dietary pulses—a 1-cup (266 g) increase in dietary pulses was associated with a reduction in HbA1c by 0.24% (−0.45, −0.03), *p* = 0.028.

### 3.4. Portfolio Diet and Other Cardiometabolic Risk Factors 

Appendix A presents the association between change in the c-PDS and change in other cardiovascular risk factors. An increase in the c-PDS was associated with a decrease in established lipid targets for CVD [7], LDL-C (β: −0.03 mmol/L per c-PDS point, 95% CI: −0.05, −0.005, *p* = 0.016), and non-HDL-C (β: −0.04 mmol/L per c-PDS point, 95% CI: −0.06, −0.02, *p* = 0.001). Additionally, an increase in the c-PDS was associated with a decrease in total cholesterol (β: −0.04 mmol/L per c-PDS point, 95% CI: −0.06, −0.01, *p* = 0.002) and triglycerides (β: −0.04 mmol/L per c-PDS point, 95% CI: −0.07, −0.007, *p* = 0.017). No association was observed for HDL-C or systolic and diastolic blood pressure (*p* > 0.05). 

### 3.5. Sensitivity and Subgroup Analyses

Appendix A presents the scatter plot of the change in the c-PDS and the change in GI. Change in the c-PDS was negatively correlated with change in GI over the 6 months (r = −0.55, *p* < 0.001). Appendix A presents the mediation analyses with dietary GIs and changes in body weight. When dietary GI was included as a covariate in the multiple linear regression model of the c-PDS and HbA1c, the association was attenuated—the beta coefficient was halved and the modeled association was no longer significant (β: −0.02% per c-PDS point, 95% CI: −0.05, 0.01, *p* = 0.287). For body weight change, there was attenuation of the association between the Portfolio Diet and HbA1c; however, the association remained significant (β: −0.03% per c-PDS point, 95% CI: −0.05, −0.01, *p* = 0.005). Appendix A presents the stratified analysis for the association between change in the c-PDS and change in HbA1c by sex and ethnicity. There was no effect modification by either sex or ethnicity (*p*_-interaction_ > 0.05).

## 4. Discussion

### 4.1. Principle Findings

This patient-level pooled analysis of two randomized dietary trials assessed the association of their c-PDS with changes in HbA1c over a 6-month period in older, predominantly White adults living with T2DM. Our findings showed that greater adherence to the Portfolio Diet was associated with lower HbA1c, where a clinically meaningful 0.32% (95% CI: −0.51 to −0.13) lower HbA1c may be observed with a 7.5-point (30%) increase in the c-PDS [37,38,39]. The reduction in HbA1c seen for the total c-PDS may be driven by changes in intake of nuts and seeds and plant protein, where increased points from each were associated with significant reductions in HbA1c. A sub-analysis of plant protein intake designed to explore dietary pulses, a particularly low-GI food, revealed an increase in dietary pulse intake was significantly associated with a reduction in HbA1c, where a reduction of −0.24% may be observed per 1 cup (226 g) of cooked pulses per day, which corresponds to 2 Portfolio points (Table 3).

Adherence to the Portfolio Diet increased from 16% to 18% at 6 months. While the pooled adherence was low, there was a large range of adherence across the participants, ranging from 0 to 60% and from 0 to 74% at baseline and at 6 months, respectively. Furthermore, when separated by original trial intervention, the 6-month c-PDS for those in the control groups was 2.6 points (10%), ranging from 0 to 13 points (0% to 52%), and for those in the test groups was 5.8 points (23%), ranging from 0.5 to 18.3 points (2% to 74%). Also, as Appendix A presents, adherence generally decreased in the control group (high fibre dietary advice) and increased in the test (low GI dietary advice) group, as expected. The mean change in the c-PDS from baseline to 6 months for those in the control groups was −1.5 points, ranging from −7.8 to 7.4 points. The mean change for those in the test groups was 1.7 points, ranging from −6.3 to 9.4 points. The little to no change in intake of the individual Portfolio Diet pillars over the trial period (Appendix A) may explain the lack of associations found for certain pillars. Viscous fibre may also be of interest, as there was a pattern tending towards significance. Beyond HbA1c, greater adherence to the Portfolio Diet was also associated with lower LDL-C and non-HDL-C (Appendix A), which are established lipid targets for CVD (7). These findings were anticipated given that the diet was designed to target LDL-C directly (15). The Portfolio Diet was also inversely associated with other cardiometabolic measures (total cholesterol and triglycerides); however, no association with HDL-C or blood pressure was observed. 

We also observed a negative correlation between the c-PDS and GI, suggesting that as adherence to the Portfolio Diet increases the GI of the diet decreases. This correlation between the two diets was expected due to the overlap in some of the prescribed foods, such as dietary pulses. We demonstrated that inclusion of dietary GI within the statistical model attenuated the association between the Portfolio Diet (exposure) and HbA1c (outcome). Therefore, as predicted, dietary GI may be a key mediator and mechanism for how the Portfolio Diet may improve HbA1c. Importantly, although significant, the correlation between the Portfolio Diet and the GI was not substantial (r = 0.55), indicating that while these dietary patterns share common foods, there are some differences. A future trial demonstrating the direct causal effect of the Portfolio Diet, with greater adherence (beyond the 18% found in the present dataset) including across its five pillars, will be of interest to determine the greater effectiveness of the Portfolio Diet for diabetes management. In the meantime, the present findings suggest that since both original GI dietary trials demonstrated significant improvements in glycemic control and the present data show similar results, multiple dietary patterns including the Portfolio Diet may be recommended for diabetes management.

When considering the potential mechanisms through which the Portfolio Diet may work to improve HbA1c, the displacement of high GI carbohydrates foods with low GI foods such as nuts, beans, lentils, etc., has been previously observed to improve cardiometabolic outcomes [40,41]. In addition to dietary GI, viscous fibre is believed to delay gastric emptying, increasing satiety and decreasing energy intake [42]. As half of the participants (48%) included in our analysis were considered obese, the potential role of dietary interventions on satiety, which may influence adipose tissue distribution and improve glycemic control, is an important mechanism to consider. In the DM-MRI study, change in HbA1c was associated with change in body weight [32]. In our mediation analysis, we observed that while body weight attenuated the association between the Portfolio Diet and HbA1c by 25%, the association remained significant (Appendix A). This finding suggests that while a reduction in body weight may explain some of the associated benefits of the Portfolio Diet on HbA1c, additional mechanisms drive this association.

Additionally, short-chain fatty acids (SCFA), produced by the fermentation of fibre-rich foods by bacteria in the colon, may inhibit fatty acid lipolysis—reducing levels of circulating free fatty acids—and may improve tissue insulin sensitivity [43,44]. Inflammation may also be an important mechanism, as the Portfolio Diet has previously been observed to improve CRP [16]. These observed anti-inflammatory effects of the diet may be related to an increase in dietary polyphenols from nuts and dietary pulses [45,46] which may have downstream influences on chronic diseases [47]. Moreso, the replacement of carbohydrates with MUFA has also been observed to improve markers of inflammation [48].

While no published trials of the Portfolio Diet have been conducted in a population living with T2DM and with assessed measures of glycemic control [16], there is epidemiological evidence that supports the present findings. In Spanish adults from the Prevención con Dieta Mediterránea (PREDIMED)-PLUS trial, the Portfolio Diet measured using the Portfolio Diet Score (PDS)—a validated population-dependant score for assessing the Portfolio Diet from food frequency questionnaires in epidemiological settings [49]—was associated with improved measures of glycemic control over 1 year, with a 1-SD increase in the PDS being significantly associated with a lower HbA1c of −0.02% and a fasting glucose of −0.47 mg/dL [17]. Additionally, among U.S. adult women from the Women’s Health Initiative, the Portfolio Diet was associated with a 23% lower risk of T2DM incidence when comparing highest to lowest quintiles of adherence [18].

Plant-based dietary patterns have been associated with long-term health benefits for chronic diseases in cohort studies, with evidence for reductions in T2DM incidence [50]. Importantly, these observed benefits on chronic diseases are strengthened when certain healthy plant-based foods (dietary pulses, nuts, fruits and vegetables, whole grains, and healthy oils) are emphasized [50]. The Portfolio Diet shares many similar foods with a healthy plant-based dietary pattern, while having a particular focus on cholesterol-lowering foods. Encouraging healthy, fortified, plant-based foods may be important, as it has been observed that adults consuming a plant-based diet generally have lower protein, vitamin B12, vitamin D, iron, zinc, iodine, and calcium intakes [51], which are nutrients of concern in Canadians [52]. Other dietary patterns which share similar foods to the Portfolio Diet, including the Dietary Approach to Stop Hypertension (DASH) diet and Mediterranean diet, have been observed to improve measures of glycemic control. A systematic review and meta-analysis of the DASH diet and glycemic control included two controlled trials (one in participants living with T2DM and one in those with gestational diabetes) and demonstrated a reduction in HbA1c of 0.53% [53]. The latest systematic review and meta-analysis of the Mediterranean diet and glycemic control in those living with T2DM included seven RCTs and demonstrated a reduction in HbA1c of 0.39% [54]. While no subgroup differences were observed for sex—and in the original Portfolio Diet trials no effect differences by sex were reported [15,48,55]—sex-differences have been observed in nutrition research where lower carbohydrate quality (higher GI and glycemic load) has been associated with a stronger risk of T2DM and CVD in females compared to males [56,57]. Furthermore, although no interaction based on ethnicity was observed, given the relatively small sample of South Asian and East Asian groups and the limited inclusion of other ethnicities, future analyses are necessary in diverse populations. Overall, to determine whether the effect of nutrition therapy interventions differ among certain subgroups of adults living with T2DM, additional research is needed to examine differences by sex, gender, and ethnicity, among other social determinants of health. Future research including these populations will have important clinical implications, informing clinical practice guidelines. 

### 4.2. Implications

Findings from this patient-level pooled analysis of two randomized dietary trials have important clinical implications for medical nutrition therapies and disease prevention and management. There is a need for multi-morbidity approaches for T2DM management given the overlapping risk factors it shares with other prevalent diseases, including obesity and CVD. Moreso, CVD risk is of particular concern for patients living with T2DM. CVD stands as the primary cause of mortality among individuals living with T2DM, accounting for an estimated 50% of deaths in this population [58]. Therefore, a nutrition therapy intervention that may improve hyperglycemia—in addition to having established benefits on dyslipidemia [16]—would be valuable, particularly for patients with comorbidities. 

Importantly, the level of adherence to the Portfolio Diet needed for a clinically meaningful improvement in HbA1c (0.3%) was estimated to be a 7.5-point increase (out of 25 points) in the c-PDS, corresponding to a 30% adherence to the Portfolio Diet. Given the baseline of about 4.1 c-PDS points (16% adherence), an increase of 7.5 points would result in a c-PDS of 11.6 (46% adherence). This level of adherence has been previously achieved in clinical trials of the Portfolio Diet where the intervention was dietary advice, with an average adherence of 48% in an ad libitum, multi-center trial [55]. Furthermore, given the 0.23% lower HbA1c associated per 1-cup increase in dietary pulses per day, chickpeas, beans, and lentils may be particular foods to encourage to support glycemic control as part of the plant protein pillar of the Portfolio Diet.

### 4.3. Strengths and Limitations

Strengths of the present study include, for one, the use of the c-PDS to predict concomitant changes in HbA1c as a measure of glycemic control in adults living with T2DM. HbA1c has been extensively studied as a strong indicator of T2DM incidence risk and provides a clinical measure of long-term (2–3 months) glycemic control [59]. In patients living with T2DM but without a history of CVD, when HbA1c is poorly controlled (8–8.9%) their risk of CVD mortality increases compared to that of patients with established CVD [60]. Another strength was the use of multiple linear regression analysis, which allows for possible confounders to be accounted for and for predictions about health outcomes to be made. Furthermore, analyses were performed in a population of adults living with T2DM, strengthening confidence in the estimated benefit for T2DM management. 

Nonetheless, this study has some limitations. This was a secondary analysis of data from two randomized controlled trials, and, therefore, no direct assessment of benefit to participants can be made. However, this study strengthens the rationale for a randomized controlled trial demonstrating the direct causal effect of the Portfolio Diet on measures of glycemic control in adults living with T2DM. Residual confounding is an additional limitation; however, we did adjust our multilinear regression analyses for several potential confounding variables. We also attempted to explore factors which may influence the results, including mediation analyses for dietary GI and for change in body weight, both of which attenuated the association. However, the possibility of residual confounding, including unknown factors, cannot be ruled out. Furthermore, while this study was undertaken in adults living with T2DM, the sample was limited in diversity, with an unequal representation of women and non-White groups, limiting the generalizability to other ethnicities. There was also an emphasis on only some of the Portfolio Diet foods. The relationship was predominantly driven by two pillars: nuts and seeds and plant protein, including dietary pulses. There was a low intake of plant sterols and MUFA oils, limiting our ability to assess these foods association with glycemic control in the present study.

## 5. Conclusions

In this analysis of 276 adults living with T2DM, the Portfolio Diet was associated with a reduction in HbA1c in older, predominantly White adults over a 6-month period. The relationship was driven by two pillars: nuts and seeds and plant protein, including dietary pulses. These data have implications for considering the Portfolio Diet in dietary recommendations for glycemic control in adults living with T2DM. However, data from randomized controlled trials demonstrating the direct causal effect of the Portfolio Diet on measures of glycemic control in adults living with T2DM are warranted. Analyses of these data by sex, gender, and socio-cultural related risk factors will be particularly important for understanding the translation of the Portfolio Diet across different populations.

## Figures and Tables

**Figure 1 nutrients-16-02817-f001:**
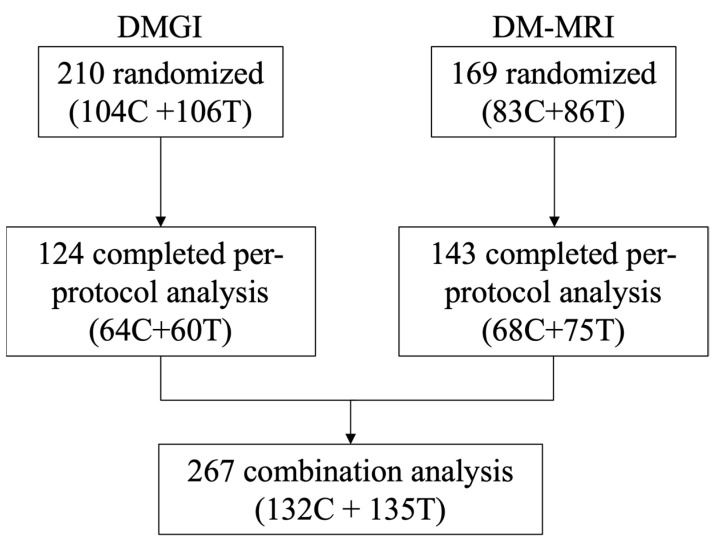
Participant flow from both trials. Dietary and HbA1c data were available at baseline and 6 months in 267 participants. Abbreviations: C, Control arm (high fibre dietary advice); DMGI, Diabetes Management with a low Glycemic Index diet study; DM-MRI, DM-Magnet Residence Imagining study; T, Test arm (low glycemic index dietary advice).

**Figure 2 nutrients-16-02817-f002:**
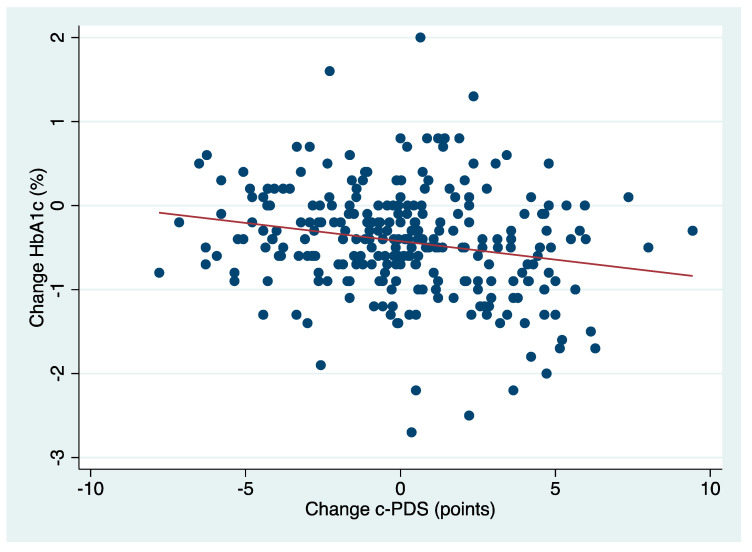
Correlation of change in the c-PDS and change in HbA1c. β: −0.04% per c-PDS point, 95% CI: −0.07 to −0.02, *p* = 0.001. Abbreviations: CI, confidence interval; c-PDS, clinical Portfolio Diet Score; HbA1c, hemoglobin A1c.

**Table 1 nutrients-16-02817-t001:** Baseline characteristics of participants in the included trials.

Characteristic	Pooled	DMGI	DM-MRI	*p* Value *
No.	267	124	143	
Age, years	61.9 (0.5)	61.8 (0.9)	62.1 (0.7)	0.76
Sex, females, *n* (%)	98 (36.7)	45 (36.3)	53 (37.1)	0.90
T2DM	267 (100)	124 (100)	143 (100)	1.00
Body weight, kg	85.5 (1.07)	86.4 (1.72)	84.8 (1.33)	0.46
BMI, kg/m^2^	30.2 (0.3)	30.5 (0.5)	30.0 (0.4)	0.50
HbA1c, %	7.08 (0.03)	7.06 (0.05)	7.10 (0.05)	0.59
TC, mmol/L	4.1 (0.06)	4.2 (0.08)	3.9 (0.09)	0.03
HDL-C, mmol/L	1.1 (0.02)	1.1 (0.02)	1.1 (0.02)	0.25
TG, mmol/L	1.5 (0.05)	1.4 (0.07)	1.5 (0.08)	0.43
LDL-C, mmol/L	2.3 (0.05)	2.5 (0.07)	2.2 (0.07)	0.001
Non-HDL-C, mmol/L	2.9 (0.06)	3.1 (0.07)	2.8 (0.08)	0.007
Smoking, *n* (%)	15 (5.6)	8 (6.5)	7 (4.9)	0.58
CVD event, *n* (%)	19 (7.1)	9 (7.3)	10 (7.0)	0.93
Ethnicity, *n* (%)				0.15
White	180 (67.4)	87 (70.2)	93 (65.0)	
South Asian	43 (16.1)	18 (14.5)	25 (17.5)	
East Asian	17 (6.4)	8 (6.5)	9 (6.3)	
African	13 (4.9)	7 (5.6)	6 (4.2)	
Hispanic	5 (1.9)	3 (2.4)	2 (1.4)	
Native	1 (0.4)	1 (0.8)	0	
Other	8 (3.0)	0	8 (5.9)	
Dietary GI	56.7	57.8	55.8	0.001
Medication use, *n* (%)				
Glycemic-control	264 (99)	122 (98)	142 (99)	0.48
Thiazolinedione	50 (19)	42 (34)	8 (6)	0.001
Biguanide	232 (87)	99 (80)	133 (93)	0.001
Sulfonylureas	109 (41)	61 (49)	48 (34)	0.01
Meglitinides	6 (2)	3 (2)	3 (2)	0.86
α-glucosidase inhibitors	6 (2)	5 (4)	1 (1)	0.07
DPP-4 inhibitors	29 (11)	0	29 (20)	0.001
Lipid-lowering	186 (70)	85 (69)	101 (71)	0.71
Statin	182 (68)	81 (65)	101 (71)	0.35
Non-statin	15 (6)	6 (5)	9 (6)	0.61
Blood pressure	186 (70)	85 (69)	101 (71)	0.71
Diuretic	51 (19)	27 (22)	24 (17)	0.30
ACEi/ARBs	168 (63)	79 (64)	89 (62)	0.80
Beta blockers	24 (9)	8 (6)	16 (11)	0.18
Calcium channel blockers	46 (17)	16 (13)	30 (21)	0.08
Alpha blockers	3 (1)	2 (2)	1 (1)	0.48
c-PDS	4.1 (0.3)	4.2 (0.3)	4.0 (0.2)	0.76

Abbreviations: ACEi, angiotensin-converting enzyme inhibitors; ARBs, angiotensin II receptor blockers; BMI, body mass index; c-PDS, clinical Portfolio Diet Score; CVD, cardiovascular disease; DDP-4, dipeptidyl peptidase 4; DMGI, Diabetes Management with a low Glycemic Index diet study; DM-MRI, DM-Magnet Residence Imagining study; GI, glycemic index; HbA1c, hemoglobin A1c; HDL-C, high-density lipoprotein-cholesterol; *n*, number; SE, standard error; TC, total cholesterol; TG, triglycerides; T2DM, type 2 diabetes mellitus. Means (SE), unless otherwise noted. * *p* value for difference across study groups.

**Table 2 nutrients-16-02817-t002:** Pooled baseline intake of the Portfolio Diet measured by the c-PDS at baseline and end (6 months) along with the corresponding change in the pooled trials.

Portfolio Diet	Baseline *	End (6 Months) *	Change at 6 Months *
c-PDS	4.10 (0.17)	4.44 (0.18)	0.13 (0.19)
Nuts and seeds	1.34 (0.08)	1.08 (0.08)	−0.27 (0.09)
Plant protein	0.51 (0.04)	0.76 (0.06)	0.25 (0.06)
Viscous fibre	0.82 (0.06)	0.97 (0.07)	0.15 (0.08)
Plant sterols	0.05 (0.02)	0.05 (0.02)	0.00 (0.02)
MUFA oils	1.37 (0.08)	1.37 (0.08)	−0.01 (0.09)
Dietary pulses, g/d	31.0 (4.20)	49.0 (4.70)	18.1 (5.90)

Abbreviations: c-PDS, clinical Portfolio Diet Score; MUFA, monounsaturated fatty acids; SE, standard error. * Values are means (SE) and presented as points allotted to the c-PDS, unless otherwise noted. To convert to % adherence, divide points by total score. For total c-PDS, divide by 25 points. For individual pillars, divide by 5 points.

**Table 3 nutrients-16-02817-t003:** Multiple linear regression analysis of associated change of the c-PDS and HbA1c.

Portfolio Diet	β Coefficient *	95% CI	*p*-Value
**c-PDS ****	−0.04	(−0.07, −0.02)	0.001
Nuts and seeds	−0.07	(−0.12, −0.02)	0.009
Plant protein	−0.11	(−0.18, −0.03)	0.009
Viscous fibre	−0.05	(−0.11, 0.01)	0.09
Plant sterols	−0.30	(−0.64, 0.04)	0.08
MUFA oils	−0.02	(−0.08, 0.03)	0.44
Dietary pulses, (per 1 cup, 266 g/d)	−0.24	(−0.45, −0.03)	0.028

Abbreviations: CI, confidence interval; c-PDS, clinical Portfolio Diet Score; HbA1c, hemoglobin A1c; MUFA, monounsaturated fatty acid. * Adjusted for age (continuous), sex (female, male), smoking (yes, no), body mass index (BMI) (continuous), prior CVD event (yes, no), antihyperglycemic medications (yes, no), blood pressure medications (yes, no), lipid-lowering medications (yes, no), and trial (DMGI, DM-MRI). ** Change per 1 point (unless otherwise indicated). *p* value for linear regression.

## Data Availability

Data is contained within the article.

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
