# Peer review of "The Portfolio Diet and HbA1c in Adults Living with Type 2 Diabetes Mellitus: A Patient-Level Pooled Analysis of Two Randomized Dietary Trials"

_nutrients, 2024, doi:10.3390/nu16172817_

Round 1

Reviewer 1 Report

Comments and Suggestions for Authors

GENERAL COMMENTS

The findings further add to the body of knowledge on the topic of a specific dietary pattern, the portfolio diet, and its impact on health, in this case on HbA1c in adults with T2D. The manangement of patients with T2D is not easy only with simple lifestyle interventions. The sample size and an adequately carried out experimental design are clearly described. The authors have addressed this clinical question focusing on the changes on glycemic control over a 6 month period. Nuts, seeds and plant protein were identified as the pillars of the dietary pattern.

Some suggestions are provided that may further improve the manuscript.

Intro, Page 2, first sentence: suggest to add a more recent reference on the increasing prevalence of T2D like for example GBD 2021 Diabetes Collaborators. Global, regional, and national burden of diabetes from 1990 to 2021, with projections of prevalence to 2050: a systematic analysis for the Global Burden of Disease Study 2021. Lancet. 2023 Jul 15;402(10397):203-234. doi: 10.1016/S0140-6736(23)01301-6. Epub 2023 Jun 22. Erratum in: Lancet. 2023 Sep 30;402(10408):1132. doi: 10.1016/S0140-6736(23)02044-5. PMID: 37356446; PMCID: PMC10364581. // GBD 2021 Risk Factors Collaborators. Global burden and strength of evidence for 88 risk factors in 204 countries and 811 subnational locations, 1990-2021: a systematic analysis for the Global Burden of Disease Study 2021. Lancet. 2024 May 18;403(10440):2162-2203. doi: 10.1016/S0140-6736(24)00933-4. Erratum in: Lancet. 2024 Jul 20;404(10449):244. doi: 10.1016/S0140-6736(24)01458-2. PMID: 38762324; PMCID: PMC11120204.

Intro, Page 2, third sentence: suggest to add “…and excess weight” to glycemic control since both are intimately related and add a recent supporting reference like for example Perdomo CM, Cohen RV, Sumithran P, et al. Contemporary medical, device, and surgical therapies for obesity in adults. Lancet. 2023;401(10382):1116-30.).

Intro, Page 2, end of the section: formulate the specific working hypothesis (what was expected and why) before stating the aims.

Page 3, line 125: provide full details for reference #40 (currently, in the reference section it only states the year – 2015.

Page 3, line 129: include “at” before the temperature.

Page 4, Table 1: do the authors have information on waist circumference and/or waist-to height ratio, which are useful tools of fat distribution and have been linked to glycemic control?

Page 5, line 178: the adherence to the dietary pattern seems quite low (16 and 18%); this needs to be commented and explained here or in the Discussion.

Page 5, Results: in order to be able to better identify the effect of the Portfolio Diet it would be useful to show the same variables listed in  Table 1 but after the 6 month intervention period.

Discussion: plant-based diets are associated with good health and are also recommended for environmental sustainability. However, there is a wide range of plant-based definitions ranging from the traditional vegetarian diets (including vegan) to semi-vegetarian/flexitarian diets. It would be also interesting to comment and cite more long-term changes to highlight that those consuming these type of diets are more likely to meet recommended intakes for carbohydrate, dietary fibre and vitamin E and are less likely to meet recommendations for protein, vitamin B12 and iodine compared to omnivores (supporting reference like for example Kent G, Kehoe L, Flynn A, Walton J. Plant-based diets: a review of the definitions and nutritional role in the adult diet. Proc Nutr Soc. 2022 Mar;81(1):62-74. doi: 10.1017/S0029665121003839. Epub 2021 Dec 20. PMID: 35760588.). Reportedly, regardless of consumer type, plant-based consumers do not meet recommendations for intakes of vitamin D, calcium and sodium. While intakes of protein, n-3, iron and zinc were generally sufficient from plant-based diets, it is important to acknowledge the lower bioavailability of these nutrients from plant-based foods compared to animal-derived products. As dietary patterns shift towards more plant-based diets, there is a need for further studies to investigate their role for nutritional adequacy and status in populations currently accustomed to consuming a primarily omnivorous diet.

Discussion: BMI data shown in Table 1 show that participants were in the obesity range; this needs to be discussed and also if significant changes in weight and fat distribution (waist or waist-to-height ratio) after the intervention took place given that the impact on adipose tissue may in part underlie the improvement seen in glycemic control (ref Heinonen S, Saarinen L, Naukkarinen J, et al. Adipocyte morphology and implications for metabolic derangements in acquired obesity. Int J Obes (Lond). 2014 Nov;38(11):1423-31. doi: 10.1038/ijo.2014.31. Epub 2014 Feb 19. PMID: 24549139).

In the same line, it might be worthwhile commenting on the potential influence of plant-based diets on inflammation and its reduction following the dietary intervention (e.g. Shanmugam G. Polyphenols: potent protectors against chronic diseases. Nat Prod Res. 2024 Aug 2:1-3. doi: 10.1080/14786419.2024.2386402. Epub ahead of print. PMID: 39092514).

Comments on the Quality of English Language

Some sentences may need to be revised for clarity.

Author Response

We thank the reviewer for their insightful comments. Please see the attachment. 

Reviewer 2 Report

Comments and Suggestions for Authors

This manuscript attempts to explore the relationship between adherence to the Portfolio Diet and the reduction of HbA1c levels in adults living with type 2 diabetes mellitus (T2DM) by pooling patient-level data from two randomized dietary trials. While the research topic is relevant to diabetes management, the manuscript falls short in several key areas that must be addressed to meet the standards expected in this field.

1. Introduction

The manuscript lacks a clear articulation of the research hypothesis and objectives. The authors mention the aim but fail to provide a concise statement of the hypothesis being tested. This omission significantly undermines the focus and readability of the paper. The authors must explicitly state the hypothesis and clarify the objectives to ensure that the study's direction is evident from the outset.

2. Methods

Given the centrality of the Portfolio Diet to this study, the current level of detail provided about this diet in the main text is insufficient. While references are cited, the manuscript itself must contain a thorough and self-sufficient explanation of the Portfolio Diet. Relying on external references is inadequate and diminishes the manuscript's comprehensiveness.

Moreover, the lack of sensitivity analyses is a glaring oversight. Without these analyses, the robustness of the findings remains questionable. The authors must conduct and present sensitivity analyses to strengthen the validity of their results.

3. Results

The manuscript reports that adherence to the Portfolio Diet was generally low, yet it proceeds to draw conclusions about the diet's efficacy. This is a fundamental flaw in the study design and interpretation. The low adherence level raises serious doubts about whether the findings can truly reflect the diet's effect. This issue is the most significant flaw in the paper and calls into question the study's overall validity.

Furthermore, the authors have neglected to analyze the characteristics of participants who did not adhere to the diet. This omission prevents a full understanding of how non-adherence may have skewed the results. A detailed analysis of these participants is essential and must be included.

4. Discussion

The discussion on the association between the Portfolio Diet and HbA1c reduction is superficial at best. The manuscript would benefit from a much deeper exploration of the biological mechanisms that might explain these effects. The current discussion lacks depth and fails to provide the necessary insights into the study's findings.

Additionally, the discussion of confounding factors is inadequate. The authors must address any residual confounding factors that could still be influencing the results. Ignoring these potential influences weakens the credibility of the study's conclusions.

Author Response

(The authors gave the same response as above.)

Round 2

Reviewer 2 Report

Comments and Suggestions for Authors

Revised manuscript is appropriate.